# First law and quantum correction for holographic entanglement contour

**Muxin Han[1,2] and Qiang Wen[3,4⋆]**

**1** Department of Physics, Florida Atlantic University,
777 Glades Road, Boca Raton, FL 33431-0991, USA
**2** Institut für Quantengravitation, Universität Erlangen-Nürnberg,
Staudtstr. 7/B2, 91058 Erlangen, Germany
**3** Shing-Tung Yau Center of Southeast University, Nanjing 210096, China
**4** School of Mathematics, Southeast University, Nanjing 211189, China

⋆ wenqiang@seu.edu.cn

## Abstract

Entanglement entropy satisfies a first law-like relation, which equates the first order perturbation of the entanglement entropy for the region $A$ to the first order perturbation of the expectation value of the modular Hamiltonian, $\delta S_A = \delta \langle K_A \rangle$. We propose that this relation has a finer version which states that, the first order perturbation of the entanglement contour equals to the first order perturbation of the contour of the modular Hamiltonian, i.e. $\delta s_A(\mathbf{x}) = \delta \langle k_A(\mathbf{x}) \rangle$. Here the contour functions $s_A(\mathbf{x})$ and $k_A(\mathbf{x})$ capture the contribution from the degrees of freedom at x to $S_A$ and $K_A$ respectively. In some simple cases $k_A(\mathbf{x})$ is determined by the stress tensor. We also evaluate the quantum correction to the entanglement contour using the fine structure of the entanglement wedge and the additive linear combination (ALC) proposal for partial entanglement entropy (PEE) respectively. The fine structure picture shows that, the quantum correction to the boundary PEE can be identified as a bulk PEE of certain bulk region. While the *ALC proposal* shows that the quantum correction to the boundary PEE comes from the linear combination of bulk entanglement entropy. We focus on holographic theories with local modular Hamiltonian and configurations of quantum field theories where the *ALC proposal* applies.

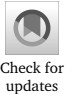

# 1   Introduction

The entanglement entropy captures the quantum entanglement in a pure state between $A$ and $B$ for a bipartite system $A \cup B$. The study of entanglement entropy has played an essential role in our understanding of the emergence of spacetime and holography. These progresses begin with the Ryu-Takayanagi (RT) proposal [1, 2] that reveals the deep connection between the spacetime geometry and quantum entanglement. In AdS/CFT [3–5], consider a static region $A$ in the boundary field theory and the minimal surface $\mathcal{E}_A$ that is in the dual AdS bulk and anchored to $\partial A$, the RT formula relates the entanglement entropy of $A$ to the area of $\mathcal{E}_A$ in Planck units, i.e.

$$S_A = \frac{\text{Area}(\mathcal{E}_A)}{4G} \,. \tag{1}$$

This relation between the quantum entanglement and geometry has recently been extended to holographic theories beyond AdS/CFT, for example the (warped) AdS/(warped) CFT correspondence [6–9] and 3-dimensional flat holography [10–12], whose dual field theory is non-Lorentz invariant. These new relations are derived firstly via the Rindler method [13], which constructs a Rindler transformation that maps the entanglement wedge to a Rindler spacetime with infinitely far away boundaries, then calculates the entanglement entropy via the thermal entropy in the Rindler spacetime. Later they are also derived in [14] via the Lewkowycz-Maldacena prescription [15, 16], which directly applies the replica trick in the bulk to calculate the entanglement entropy. However, in these cases, there exists a subtle issue about the cut-off in the bulk causing the RT surfaces not anchored on the boundary[1]. This issue can be solved at least in 2+1 dimensions by introducing certain null geodesics emanating from the boundary of $A$. Indeed the analogue of the RT surface $\mathcal{E}_A$ is the extremal geodesic whose length is at the saddle among all the geodesics that anchored on the null geodesics, then the holographic entanglement entropy is given by the length of the extremal geodesic.

These novel null geodesics in holographic theories with non-Lorentz invariant duals are indeed ingredients of the entanglement wedge's fine structure based on the bulk and boundary modular flows [17]. The fine structure also largely inspires the following study on the entanglement contour or the partial entanglement entropy [17–21]. For a given region $A$ and a subset $A_i$ of $A$, the partial entanglement entropy (PEE), denoted by $s_A(A_i)$, is defined to capture the contribution from $A_i$ to the entanglement entropy $S_A$. The key property featured by the PEE is the additivity, which is not possessed by any other entanglement measures. When the subsets reduce to single points in $A$, the PEE reduces to a function $f_A(\mathbf{x})$ called the entanglement contour [18]. $f_A(\mathbf{x})$ gives the contribution from the site at the position $\mathbf{x}$ in $A$ to $S_A$, in

---

[1]When the boundary field theory is non-Lorentz invariant, the causal development of an interval becomes an infinitely long strip instead of a causal diamond. In order to keep the consistency between the bulk and boundary causal structure, the extremal surface should not be anchored on the boundary. See [14] for more details.

other words, it is the density function of the entanglement entropy $S_A$,

$$S_A = \int_A f_A(\mathbf{x})d^d x, \qquad (2)$$

where $d$ is the dimension of $A$. The PEE $s_A(A_i)$ can also be written as

$$s_A(A_i) = \int_{A_i} f_A(\mathbf{x})d^d x, \qquad (3)$$

hence, only collect the contribution in the subset $A_i$.

Though the definition of PEE based on the reduced density matrix is still missing, the physical meaning as the density function for the entanglement entropy requires the PEE to satisfy the following physical requirements [2]:

1. *Additivity*: If $A_i^a \cup A_i^b = A_i$ and $A_i^a \cap A_i^b = \emptyset$, by definition we have

$$s_A(A_i) = s_A(A_i^a) + s_A(A_i^b). \qquad (4)$$

2. *Invariance under local unitary transformations*: $s_A(A_i)$ is invariant by any local unitary transformation inside $A_i$ or $\bar{A}$.

3. *Symmetry*: For any symmetry transformation $\mathcal{T}$ under which $\mathcal{T}A = A'$ and $\mathcal{T}A_i = A_i'$, we have

$$s_A(A_i) = s_{A'}(A_i'). \qquad (5)$$

4. *Normalization*: $S_A = s_A(A_i)|_{A_i \to A}$.

5. *Positivity*: $s_A(A_i) \geq 0$.

6. *Upper bound*: $s_A(A_i) \leq S_{A_i}$.

7. *Symmetry under the permutation*: $\mathcal{I}(\bar{A}, A_i) = s_A(A_i) = s_{\bar{A}_i}(\bar{A}) = \mathcal{I}(A_i, \bar{A})$.

There have been four PEE (or entanglement contour) proposals that satisfies the above requirements. The first one is the Gaussian formula [18, 22–28] that applies to the Gaussian states in free theories. The second proposal is a geometric construction [14, 17, 29] in holographic theories, based on the fine structure analysis of the entanglement wedge following the boundary and bulk modular flows. The third one, previously given by the author in [17, 20], claims that the PEE is given by an additive linear combination of subset entanglement entropies. Later we will call this proposal the ALC (additive linear combination) proposal for short[3]. The fourth proposal [21] follows the construction of the extensive (or additive) mutual information (EMI) [31] (see also [32] for a similar construction), which tried to solve the above seven requirements in CFT. The entanglement contour can also be studied under the picture of the bit threads [33] in holographic theories, see for example [19, 34–36]. The PEE calculated by different approaches are highly consistent with one another [14, 17, 21, 28, 29], suggesting that the PEE should be well-defined and unique. The uniqueness of the PEE has been confirmed for Poincaré invariant theories [21], by showing that the above seven requirements in these theories have unique solution. The PEE is also useful to study the entanglement

---

[2]The requirements 1-6 are firstly given in [18], while the requirement 7 is recently given in [21]

[3]Previously in [20, 21, 30], this proposal was call the "*partial entanglement entropy proposal*". This is a bit misleading since we defined the PEE as (3) rather than the linear combination (6).

structure in condensed matter theories[4]. Recently, the entanglement contour is used to give the entanglement structure of the Hawking Radiation which shows non-trivial behavior [41, 42] due to an island phase transition (see [43] for a review on this topic). The above progresses suggest that the new concept of entanglement contour in quantum information should play an important role in our understanding of the gauge/gravity duality and the entanglement structure in quantum field theories (or many-body system).

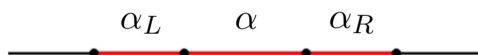

Figure 1: A typical region $A$ with a definite order is shown by the red interval. When an arbitrary subset $\alpha$ is chosen, a natural decomposition of $A = \alpha_L \cup \alpha \cup \alpha_R$ is determined. All the degrees of freedom in $A$ lines in a definite order. When $A$ is a circle, the definition of $\alpha_L$ and $\alpha_R$ become ambiguous.

In this paper, we mainly use the fine structure of the entanglement wedge and the *ALC proposal* to construct the PEE. The *ALC proposal* [17, 20] claims that, the PEE is given by a linear combination of certain subset entanglement entropies. The ALC proposal is proven to satisfy all the seven requirements using only the general properties of entanglement entropy. It can be applied to generic theories, but a definite order is required for all the degrees of freedom in $A$ for satisfying the additivity.

- *The ALC proposal*: Given a region $A$ and an arbitrary subset $\alpha$, when there is a definite order inside $A$, it can be unambiguously partitioned into three non-overlapping subregions $A = \alpha_L \cup \alpha \cup \alpha_R$ (see for example Fig.1), where $\alpha_L$ ($\alpha_R$) denotes the subset on the left (right) hand side of $\alpha$. In this configuration, the *ALC proposal* claims that

$$s_A(\alpha) = \frac{1}{2}\left(S_{\alpha_L \cup \alpha} + S_{\alpha \cup \alpha_R} - S_{\alpha_L} - S_{\alpha_R}\right). \tag{6}$$

The *ALC proposal* can be used to calculate the entanglement contour for one dimensional regions in general theories with a definite order [17, 20]. It also works for highly symmetric regions in higher dimensions, which can be characterized by a single coordinate [29]. Furthermore, this linear combination can be understood as a conditional mutual information [42]

$$s_A(\alpha) = \frac{1}{2}I(\alpha : \bar{A}|\alpha_L) = \frac{1}{2}I(\alpha : \bar{A}|\alpha_R). \tag{7}$$

We will briefly introduce the fine structure approach in section 3 later.

By an infinitesimal variation of the state, the perturbation of entanglement entropy $S_A$ satisfies a first law-like relation $\delta S_A = \delta\langle K_A \rangle$, where $\langle K_A \rangle$ is the expectation value of the modular

---

[4]The entanglement contour gives a finer description for the entanglement structure. In condense matter theories it can be used to discriminate between gapped systems and gapless systems with a finite number of zero modes in $d = 3$ [18]. It has been shown to be particularly useful to characterize the spreading of entanglement when studying dynamical situations [18, 19, 27]. The entanglement contour is also a useful probe of slowly scrambling and non-thermalizing dynamics for some interacting many-body systems [37] and holographic states dual to Bañados geometries, and general excited states in the small interval limit [38]. Holographically, the correspondence between PEE and bulk geodesic chords [14, 17] is a finer correspondence between the quantum entanglement and bulk geometry [14, 39]. Under some balanced condition the PEE also gives the area of the entanglement wedge cross section [30]. The balanced PEE can be considered to be an generalization of the reflected entropy [40] to generic purifications of the bipartite system [30].

Hamiltonian $K_A$ [44,45]. In holographic theories where the RT formula applies, $\delta S_A$ affects the dynamics of the bulk geometry: The first law of entanglement entropy has been used to derive the linearized Einstein's equations in the bulk spacetime [45,46]. The first law and linearized Einstein's equations have also been discussed in holographies beyond AdS/CFT [12,47,48]. In this paper we propose that the first law of entanglement entropy has a finer description: in a given region $A$, the first order perturbation of the entanglement contour at each site equals the first order variation of the expectation value of the modular Hamiltonian's contour, which is a similar density function for the modular Hamiltonian.

Another important topic of holographic entanglement entropy is the quantum correction. The RT formula only concerns the leading order contribution to the entanglement entropy $S_A$. It is shown in [49] that the quantum correction to $S_A$ is just the bulk entanglement entropy of the homology surface $\Sigma_A$ in the entanglement wedge. The evaluation of the quantum correction plays an essential role in our understanding of holography and spacetime beyond the classical level, see for example [50–55]. For holographic theories, the entanglement contour derived [14,17,29] via the fine structure of the entanglement wedge is also only at the leading order. Then it is very interesting to explore the finer description of the quantum correction. More explicitly, for a give subset $A_i$ of $A$, we want to evaluate the quantum correction to the PEE $s_A(A_i)$. Furthermore, for the cases that the modular Hamiltonian is local, we identify a bulk sub-region $a_i$ of the homology surface $\Sigma_A$, such that the contribution from $a_i$ to the bulk entanglement entropy $S_{\Sigma_A}$ gives the quantum correction to $s_A(A_i)$.

## 2 The first law of entanglement contour and the contour of modular Hamiltonian

The state of a generic quantum system can be described by the density matrix $\rho_{total}$. Let us consider an arbitrary subsystem $A$ and its complement $\bar{A}$, the state of $A$ is then described by the reduced density matrix $\rho_A = \text{Tr}_{\bar{A}}\rho_{total}$. If the total system is in a pure state, the entanglement between $A$ and $\bar{A}$ is captured by the entanglement entropy that is the von Neumann entropy $S_A$ of $\rho_A$

$$S_A = -\text{Tr}\rho_A \log \rho_A. \tag{8}$$

The modular Hamiltonian $K_A$ is a state-dependent operator defined by

$$\rho_A \equiv e^{-K_A}. \tag{9}$$

One may multiply a constant to the right hand side of the above equation to ensure $\text{Tr}\rho_A = 1$. Usually the modular Hamiltonian is non-local. For the cases where $K_A$ is local, usually it can be written as $K_A = -H/T$, where $H$ is the ordinary Hamiltonian measured by the local observer (or Rindler observer) confined in the causal development of $A$.

Let us consider any infinitesimal perturbation to the density matrix $\rho_{total}$, the first order perturbation of the entanglement entropy is given by

$$\begin{aligned}
\delta S_A &= -\text{Tr}(\delta\rho_A \log \rho_A) - \text{Tr}(\rho_A \rho_A^{-1} \delta\rho_A) \\
&= \text{Tr}(\delta\rho_A K_A) - \text{Tr}\delta\rho_A \\
&= \delta\langle K_A\rangle.
\end{aligned} \tag{10}$$

Here we have used the fact the $\text{Tr}\delta\rho_A = 0$, since $\text{Tr}\rho_A = 1$ always holds and $K_A$ is defined by the unperturbed state. The above equality between variations of the entanglement entropy and the modular Hamiltonian's expectation value is called the *first law of entanglement entropy*.

For thermal states where $K_A = -H/T$, this relation becomes the quantum version of the first law of thermodynamics, $\delta\langle H\rangle = T\delta S_A$.

It is very interesting to explore a finer version of the above first law, i.e. the relation between variations of the entanglement contour and certain local properties of modular Hamiltonian. Here we focus on the configurations where the *ALC proposal* applies. This includes single intervals in 2-dimensional theories and spherical (or strip) regions in higher dimensions with a definite order. Again, let us consider a region $A$ and its non-overlapping connected subsets $\{A_1, A_2, A_3\}$, the PEE is given by the *ALC proposal*

$$s_A(A_2) = \frac{1}{2}\left(S_{12} + S_{23} - S_1 - S_3\right). \tag{11}$$

Here we write, for example, $S_{A_1 \cup A_2} = S_{12}$. Similarly the modular Hamiltonian of $A_1 \cup A_2$ is denoted by $K_{A_1 \cup A_2} = K_{12}$.

Let us perform an infinitesimal perturbation on both sides of (11), then apply the first law to all the subset entanglement entropies on the right hand side, we get

$$\begin{aligned}
\delta s_A(A_2) &= \frac{1}{2}\left(\delta\langle K_{12}\rangle + \delta\langle K_{23}\rangle - \delta\langle K_1\rangle - \delta\langle K_3\rangle\right)\\
&= \frac{1}{2}\left(\mathrm{Tr}(\delta\rho_{12}K_{12}) + \mathrm{Tr}(\delta\rho_{23}K_{23}) - \mathrm{Tr}(\delta\rho_1 K_1) - \mathrm{Tr}(\delta\rho_3 K_3)\right).
\end{aligned} \tag{12}$$

We assume that the Hilbert space $\mathcal{H}_A$ of $A$ factorizes $\mathcal{H}_A = \mathcal{H}_{A_1} \otimes \mathcal{H}_{A_2} \otimes \mathcal{H}_{A_3}$. The modular Hamiltonian acts trivially outside the region where it is defined. So it is convenient to extend it to an operator acting on the whole region $A$, for example

$$K_{12} \equiv K_{12} \otimes I_3, \tag{13}$$

where $I_3$ is the identity operator on $\mathcal{H}_{A_3}$. This is crucial to write

$$\mathrm{Tr}(\delta\rho_{12}K_{12}) = \mathrm{Tr}(\delta\rho_A K_{12}), \tag{14}$$

where the trace on the left hand side is over $\mathcal{H}_{A_1} \otimes \mathcal{H}_{A_2}$, while on the right hand side the trace is over $\mathcal{H}_A$. We rewrite other terms similarly to obtain,

$$\delta s_A(A_2) = \frac{1}{2}\mathrm{Tr}[\delta\rho_A(K_{12} + K_{23} - K_1 - K_3)]. \tag{15}$$

Similarly we may express the PEE in terms of the modular Hamiltonians,

$$s_A(A_2) = \frac{1}{2}\mathrm{Tr}\rho_A(K_{12} + K_{23} - K_1 - K_3). \tag{16}$$

It is easy to see that, the linear combination of the modular Hamiltonians in the above equation is exactly the same as the subset entanglement entropies in the *ALC proposal*. It has been proven that this linear combination was additive. More explicitly, let us define a new non-local operator on $A$,

$$k_A(A_2) \equiv \frac{1}{2}\left(K_{12} + K_{23} - K_1 - K_3\right). \tag{17}$$

If $A_2$ is divided into two non-overlapping connected subregions $A = A_2^a \cup A_2^b$, we have

$$k_A(A_2) = k_A(A_2^a) + k_A(A_2^b). \tag{18}$$

It is natural to take $K_A \to 0$ when $A$ vanishes, hence when we take the limit $A_2 \to A, A_1 \to \emptyset, A_3 \to \emptyset$, we get the normalization property,

$$k_A(A_2)|_{A_2 \to A} = K_A. \tag{19}$$

Due to the additivity and normalization of the operator $k_A(A_2)$, we call $k_A(A_2)$ the *partial modular Hamiltonian*. Furthermore, if we know the modular Hamiltonian for all the subregions inside $A$, we can determine the contour function $k_A(\mathbf{x})$[5] for $K_A$ by taking $A_2$ to be a single site at the position $\mathbf{x} = \{t, \vec{x}\}$, hence

$$K_A = \int_A k_A(\mathbf{x}) d\vec{x} . \tag{20}$$

Similar to the entanglement contour, $k_A(\mathbf{x})$ is understood as a density function for the modular Hamiltonian $K_A$. The partial modular Hamiltonian $k_A(A_2)$ gives the contribution from the subregion $A_2$, i.e.

$$k_A(A_2) = \int_{A_2} k_A(\mathbf{x}) d\vec{x} , \tag{21}$$

where the domain of the integration is confined in $A_2$. Note that both the contour function $k_A(\mathbf{x})$ and the partial modular Hamiltonian $k_A(A_2)$ are operators defined on $A$ rather than the point $\mathbf{x}$ or the subregion $A_2$.

As a result, the equation (12) can be written as

$$\delta s_A(A_2) = \delta \langle k_A(A_2) \rangle , \tag{22}$$

which we call the *first law of partial entanglement entropy*. If we know all the partial modular Hamiltonians, we can determine the contour function hence get a finer version of the above relation

$$\delta s_A(\mathbf{x}) = \delta \langle k_A(\mathbf{x}) \rangle , \tag{23}$$

which we call the *first law of entanglement contour*. For any site $\mathbf{x}$ in $A$, the first law of entanglement contour states that the perturbation of the contribution to $S_A$ at $\mathbf{x}$ equals the perturbation of the expectation value of $k_A(\mathbf{x})$, which is the contribution to $K_A$ at $\mathbf{x}$. Though it is derived for the special cases where the *ALC proposal* applies, we conjecture it to be valid for more general configurations. We hope this can be confirmed in the future.

This finer version of the first law is useful, because the modular Hamiltonian has been extensively explored in many configurations, especially when the modular Hamiltonian is local. More importantly, the modular Hamiltonian $K_A$ is usually written as an integration over the region $A$, hence perfectly match with our introduction of the contour of the modular Hamiltonian. One simple and renowned case is the modular Hamiltonians for ball-shaped regions $A$ in $d$-dimensional CFTs. If we consider the vacuum state of the CFT and a static and ball-shaped region with radius $R$ and center position $\mathbf{x}_0 = \{t_0, \vec{x}_0\}$, then the modular Hamiltonian takes the simple form [13, 56],

$$K_A = 2\pi \int_A \frac{R^2 - |\vec{x} - \vec{x}_0|^2}{2R} T_{tt}(t_0, \vec{x}) d\vec{x} , \tag{24}$$

---

[5]Note that, one should not take the contour function $k_A(\mathbf{x})$ as a local function of $\mathbf{x}$ since it also depend on the region $A$. Also it is an operator in the sense of (17) rather than a number.

where $T_{\mu\nu}$ is the stress tensor and $\vec{x}$ is the coordinates on $A$. More generally, the modular Hamiltonian can be written in a covariant way,

$$K_A = \int_A d\Sigma \ \eta^\mu(\mathbf{x}) \ T_{\mu\nu}(\mathbf{x}) \ \xi^\nu(\mathbf{x}), \qquad (25)$$

where $d\Sigma$ is an infinitesimal volume on the spacelike co-dimension-one region $A$ with the normal vector $\eta^\mu$, while the vector field $\xi^\mu$ describes the modular flow (or geometric flow) which is generated by the modular Hamiltonian.

There are two ways to derive the modular flow. The first one relies on the construction of the Rindler transformation $R$, which is a symmetry transformation that maps the causal development $\mathcal{D}_A$ of $A$ to a Rindler spacetime with infinitely far away boundaries. The normal Hamiltonian, which generates the Rindler time translation $\partial_\tau$ in the Rindler spacetime, is mapped to the modular Hamiltonian of $A$. In other words, $\partial_\tau$ maps to the modular flow $\xi^u$ in $\mathcal{D}_A$ by the inverse Rindler transformation. However the construction of the Rindler mapping is highly non-trivial. The Rindler transformation for static balls in CFTs are constructed in [13]. While the Rindler transformation for covariant intervals in warped CFTs and BMSFTs (theories with BMS$_3$ symmetries) are constructed in [6,8] and [10]. See also [48] for a related construction of the modular Hamiltonian. The Rindler transformation can be extended into the bulk in the context of holography, hence plays an essential role to derive the geometric picture of the entanglement entropy [8,10].

Recently another way to generate the modular flow is proposed in [20] base on the properties of the PEE. The key of this approach is that the PEE should be invariant under the modular flow. The property is observed in the entanglement wedge's fine structure, which we will introduce later. Using the *ALC proposal*, it is easy to derive the orbit of the modular flow in $\mathcal{D}_A$, if we know all the entanglement entropies for sub-intervals inside $D_A$. This approach reproduces the previous results quite easily. More importantly, it does not rely on the Rindler transformations.

In the Rindler spacetime, the modular Hamiltonian is just the energy. Since the Rindler spacetime is invariant under the translation along the spacial directions, the contour function for the modular Hamiltonian should respect this symmetry, thus is a constant. Applying the inverse Rindler transformation, this flat contour maps to the contour function of the modular Hamiltonian $K_A$ in $A$. This contour function is nothing but the integrand of (24) and (25), i.e.

$$k_A(\mathbf{x}) = \eta^\mu(\mathbf{x}) \ T_{\mu\nu}(\mathbf{x}) \ \xi^\nu(\mathbf{x}). \qquad (26)$$

According to the first law of the entanglement contour, we have

$$\delta s_A(x) = \eta^\mu(\mathbf{x}) \ \xi^\nu(\mathbf{x}) \ \delta \langle T_{\mu\nu}(\mathbf{x}) \rangle, \qquad (27)$$

which states that, the first order variation of the entanglement contour relates to the first order variation of the stress tensor. Note that in the above equation we used the relation $\delta(\eta^\mu(\mathbf{x})\xi^\nu(\mathbf{x})) = 0$ because the geometry is at the saddle hence the first order perturbation of geometry vanishes. The above relation is also in some sense applied in [42] to evaluate the entanglement contour for low energy excited states of CFT near the vacuum.

However, in more generic configurations the modular Hamiltonian cannot be written as an integration like (25), hence one may worry about the validity of (20). We stress that, the reason we can write $K_A$ as an integral of $k_A(\mathbf{x})$ is the additivity of the *partial modular Hamiltonian*, which is always true when the *ALC proposal* applies. The *ALC proposal* does not select theories. For example, let us consider an interval $A$ in a 2-dimension theory which is not conformal invariant, where $K_A$ can not be written as (25). Since in this case the *ALC proposal* applies thus the *partial modular Hamiltonian* is additive, $K_A$ can still be written as (20) with $k_A(\mathbf{x})$ not directly related to the stress tensor. So (25) is not necessary for the validity of (20).

# 3 Quantum correction to holographic entanglement contour

## 3.1 Quantum correction to holographic entanglement entropy

The RT formula only gives the leading order contribution for the holographic entanglement entropy, i.e. at the order $\mathcal{O}(1/G_N) \sim \mathcal{O}(N^2)$. The Faulkner-Lewkowycz-Maldacena (FLM) formula [45] corrects the RT formula to the next order $\mathcal{O}(1)$,

$$S_A = \frac{\text{Area}(\mathcal{E}_A)}{4G_N} + S_{bulk}(\Sigma_A), \tag{28}$$

where $S_{bulk}(\Sigma_A)$ is the bulk entanglement entropy for the homology surface $\Sigma_A$, which is any Cauchy surface with its boundary satisfying $\partial\Sigma_A = A \cup \mathcal{E}_A$. Later in this paper we use the short-hand notation $\Sigma_A \equiv a$. The entanglement wedge $\mathcal{W}_A$ is the causal development of $\Sigma_A$. Note that, compared with the full expression for the quantum correction to the holographic entanglement entropy [45], Eq. (28) omitted the terms that are given by local integrals on the original minimal surface, including the terms that cancel the UV divergences of the bulk entanglement entropy.

Since the quantum correction is taken into account, the minimization on the area of the RT surface should be adjusted to the minimization of the quantum extremal surface [51],

$$S_A = \min\left(\frac{\text{Area}(\tilde{\mathcal{E}}_A)}{4G_N} + S_{bulk}(\tilde{\Sigma}_A)\right). \tag{29}$$

Accordingly the surface satisfying the minimization changes from $\mathcal{E}_A$ to $\tilde{\mathcal{E}}_A$, and $\Sigma_A$ changes to $\tilde{\Sigma}_A$. However, this difference usually only affect $S_A$ at the order $\mathcal{O}(G_N)$ [6], hence we will directly apply (28) instead of (29) to avoid unnecessary complications. Our discussion focuses on the configurations where the quantum correction is much smaller than the leading contribution from the RT formula.

The relation (28) implies an important relation between the bulk and boundary modular Hamiltonian [53],

$$K_A = \frac{\hat{\mathcal{E}}_A}{4G_N} + K_a, \tag{30}$$

where $K_a$ is the modular Hamiltonian of the bulk region $\Sigma_A$, and $\hat{\mathcal{E}}_A$ is the bulk area operator whose expectation value gives the area of the RT surface.

Then it is quite interesting to discuss the quantum corrections to the entanglement contour. Firstly we will explore the spatial distribution of the bulk entanglement entropy on the homology surface $\Sigma_A$, i.e. the entanglement contour or PEE of the bulk degrees of freedom. The essential entanglement contour that we study is the contour on the boundary region $A$, so the PEE from any bulk degrees of freedom will be assigned to the PEE of a boundary degrees of freedom as the quantum correction to the entanglement contour of $A$. Secondly, we will explore how to assign the bulk PEE to the boundary PEE. We will study the quantum correction of the contour using both the fine structure analysis with the modular slices and the *ALC proposal*.

---

[6] The difference only gives significant corrections when we approach a phase transition, where the RT surface jumps discontinuously or when the bulk entanglement entropy is comparable to the area term.

### 3.2 Quantum correction to entanglement contour from the fine structure

**Fine structure of the entanglement wedge**

In holography, when the modular Hamiltonian is local, the entanglement contour can be described by a geometric picture [7], which is constructed in a series of papers [14, 17, 20, 29]. Since the modular Hamiltonian is local thus generates a geometric modular flow, there exists a natural slicing of the entanglement wedge. More explicitly, from any point $P$ in $A$, the boundary modular flow generates an orbit, which we call the boundary modular flow curve. Then we let the points on the boundary modular flow curve flow under the bulk modular flow. Trajectories of this flow form a two dimensional bulk surface, which we call a modular slice. The causal development $\mathcal{D}_A$ is a slicing of the boundary modular flow curves. Similarly the entanglement wedge $\mathcal{W}_A$ is a slicing of the modular slices. Note that, since the boundary modular flow is also a bulk modular flow, when the boundary modular flow curve is settled exactly at the boundary, i.e. $z = 0$, its trajectory under the bulk modular flow is just itself. Here by the trajectory of the boundary modular flow curve, we mean the trajectory of the curve settled at the limit $z \to 0$ but $z \neq 0$, hence points on the curve can flow into the bulk. More explicitly, points on the curve flow into the bulk, then get to a turning point, and eventually flow back to some points on exactly the same boundary modular flow curve.

See Fig.2 for an explicit example in AdS$_3$/CFT$_2$. In the right figure the black curve is the boundary modular flow curve that passes the point $P$, the orange curves are orbits of points on the black curve under the bulk modular flow. The modular slice intersects with the RT surface $\mathcal{E}_A$ at the partner point $\tilde{P}$ of $P$. The outermost straight orange lines are the normal null geodesics emanated from $\mathcal{E}_A$, and are also bulk modular flow curves that end on the future and past tips of the causal development $\mathcal{D}_A$. $\tau_m$ denotes the Rindler time in different causal wedges, which can be covered by a single complex "time" coordinate,

$$\tau_m = \tau + \frac{m-1}{2}\pi i. \tag{31}$$

In this coordinate, the thermal circle in the entanglement wedge $\mathcal{W}_A$ is just the imaginary circle $\tau \sim \tau + 2\pi i$ of the Rindler time. The dashed line $\gamma_P$ is where the modular slice intersect with the homology surface $\Sigma_A$.

**The holographic entanglement contour from the fine structure**

The relation between the fine structure and the entanglement contour appears as we consider the replica story of a single point $P$ in $A$. When applying the replica trick, we prepare $n$ copies of the system and cut the region $A$ open for all the copies, then we glue them cyclically to form a $n$-manifold. Correspondingly, in the gravity side we cut the entanglement wedge open along any homology surface $\Sigma_A$, then glue all the copies of the bulk spacetime cyclically [15, 16].

While applying replica trick on $A$, let us focus on the replica story of a single point $P$, and see how it affects the boundary and bulk modular flow. Firstly we cut $P$ open for each copy. This cuts the modular flow curve open at $P$. Then we glue all the open curves cyclically at $P$ in each copy, hence the modular flow in the $i$th copy will flow into the $(i+1)$th copy of the curve through $P$. See Fig.3 for a simple example with $n = 2$. Here the boundary modular flow curve along $\tau_1$ contains the lower half line in the first copy and the upper half line in the second copy. Then we prepare two copies of the modular slices and see how the bulk modular flow

---

[7]This picture works also for holographic theories beyond AdS/CFT [14], for example, the (warped) AdS/ (warped) CFT correspondence and the flat holography. However, similar construction cannot be straitforwardly generalized to the cases of multi-intervals and a large enough boundary interval in the BTZ background with disconnected RT surface, since the modular flow becomes nonlocal. This is an important problem we hope to understand further in the future.

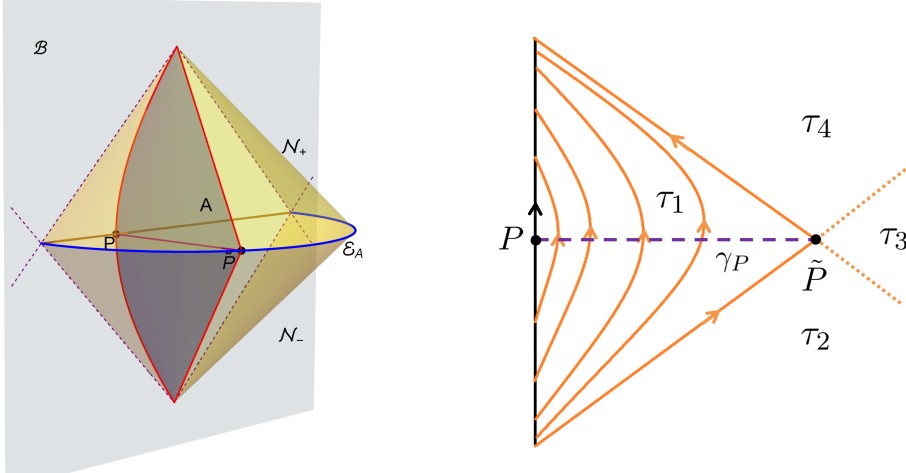

Figure 2: A typical example of the modular slice in AdS$_3$/CFT$_2$. In the left figure, the modular slice at the point $P$ is embedded in the entanglement wedge. The right figure shows how the boundary modular flow curve flows in the bulk under the bulk modular flow. The dashed curve $\gamma_P$ is where is modular slice intersect with the homology surface. $\tau_m$ denotes the modular flow in different causal wedges in the bulk. The modular flow curves become null at the boundary of the entanglement wedge, and are just the null geodesic congruence emanating from the RT surface $\mathcal{E}_A$ vertically.

is affected. Note that, the bulk modular flow lines emanating from the $\tau_1$ boundary modular flow curve should return to the same boundary modular flow curve. The fact that the $\tau_1$ boundary curve now contains two parts in different copies implies that, the $\tau_1$ bulk modular flow curves should also be cut open and glued cyclically, thus can flow back to the second part of the $\tau_1$ boundary curve in the second copy. The place where we cut the bulk modular flow curves open is just the curves $\gamma_P$, which are the purple lines in Fig.3.

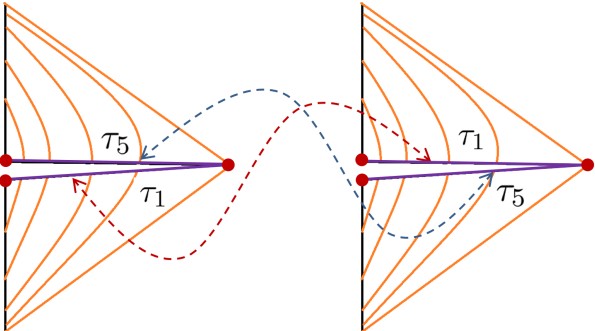

Figure 3: The replica trick applied to the modular slice when $n = 2$. Each slice is cut open at $\gamma_P$ then glued cyclically. The dashed lines show the gluing boundary conditions.

In summary, the replica story of a single point in $A$ induces the replica story of the corresponding modular slice. Since the modular flows are local, it will not affect the modular slices in the neighborhood. The replica story on all the modular slices are relatively independent and together form the replica story of the entanglement wedge. Following the calculation

of [15, 16], if we evaluate the partition functions at the classical level, the cyclic gluing at the region $A$ turns on the extensive contribution at the bulk fixed points of the replica symmetry, i.e. the RT surface. Accordingly, the cyclic gluing of any point $P$ exactly turns on the contribution to the entanglement entropy at the partner point $\tilde{P}$. This is the original statement of [17]. In the same sense this relation implies a correspondence between the geodesic chords $\mathcal{E}_i$ on $\mathcal{E}_A$ and the PEE of certain subset $A_i$ in $A$,

$$s_A(A_i) = \frac{Length(\mathcal{E}_i)}{4G}, \tag{32}$$

where $\mathcal{E}_i$ is the set of partner points of $A_i$. The one-to-one correspondence between all the points on $A$ and $\mathcal{E}_A$ gives the entanglement contour of $A$.

In the context of AdS/CFT, given a static region $A$ (spheres or intervals) and a static homology surface $\Sigma_A$, $\gamma_P$ for any point $P$ is just a static geodesic normal to $\mathcal{E}_A$ [29] [8]. See the purple dashed lines in the left figure of Fig.4. The correspondence between the PEE of the subsets $A_i$ and geodesic chords $\mathcal{E}_i$ in the sense of (32) is shown in the right figure of Fig.4. Accordingly the homology surface is also decomposed by two $\gamma_P$ curves for two points that decomposes $A$,

$$\Sigma_A \equiv a = a_1 \cup a_2 \cup a_3. \tag{33}$$

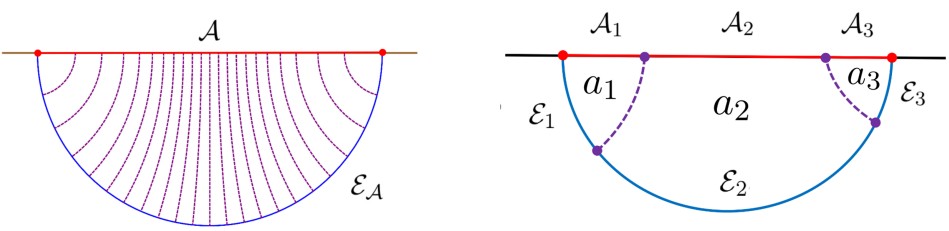

Figure 4: The above two figures show a time slice of the entanglement wedge. The purple dashed lines are the $\gamma_P$ curves, which are static geodesics normal to $\mathcal{E}_A$. In the right figure, the decomposition of $A$ induces a decomposition of the homology surface $\Sigma_A$ and the RT surface $\mathcal{E}_A$.

**Quantum correction to the holographic entanglement contour**

In the above discussion the partition functions are only evaluated at the classical level, hence the entanglement contour from the slicing of the entanglement wedge by the modular slices is only at the leading order. When including the quantum corrections to the partition functions, i.e. computing the partition function of all bulk quantum fluctuations around the classical geometry, the entanglement entropy and entanglement contour should receive quantum corrections. As we previously pointed out in section 3.1, the first order quantum correction to the entanglement entropy comes from the entanglement entropy of the bulk region in the entanglement wedge $S_a$. Studying the correction to the entanglement contour relates to the

---

[8]The curves $\gamma_P$ coincide with a special bit-thread configuration constructed in [57] following the bulk geodesics.

following question: how do we distribute the bulk entanglement entropy to the degrees of freedom in $A$?

The answer is indeed hidden in the fine structure of the entanglement wedge. Now we introduce the entanglement contour $s_a(\mathbf{x})$, which represent the contribution from the site $\mathbf{x}$ to the bulk entanglement entropy $S_a$. The cyclic gluing of the single point $P$ not only turns on the leading contribution to $S_A$ at its partner point $\tilde{P}$, but also induces the cyclic gluing of the bulk points on the curve $\gamma_P$. Note that the cyclic gluing of all the points in the homology surface $\Sigma_A$ coincides with the replica trick in the bulk for the bulk entanglement entropy $S_a$. This indicates that, the quantum correction to the PEE of the point $P$ comes from the bulk PEE of the curve $\gamma_P$, which we denote as $s_a(\gamma_P)$.

It is more convenient to consider the quantum correction to the PEE $s_A(A_2)$ of a subregion $A_2$. For example, see the left figure in Fig.5, where $A$ is a static interval which is divided into three non-overlapping parts $A = A_1 \cup A_2 \cup A_3$. According to the fine correspondence, the geodesic chord $\mathcal{E}_2$ gives the PEE $s_A(A_2)$ at the leading order. The curves $\gamma_P$ for all the points in $A_2$ form the bulk region enclosed by $A_2$, $\mathcal{E}_2$ and the $\gamma_P$ curves for the two endpoints of $A_2$. This region is denoted as $a_2$, which is the yellow region in Fig.5. In other words, we have

$$s_A(A_2) = \frac{Area(\mathcal{E}_2)}{4G} + s_a(a_2), \tag{34}$$

where the bulk PEE $s_a(a_2)$ is the quantum correction to the PEE $s_A(A_2)$.

The above statement can be easily understood in the Rindler bulk spacetime. The Rindler transformations map $\mathcal{W}_A$ to the Rindler bulk spacetime, which is an AdS black brane with translation symmetries along the directions, say $\vec{x} = \{x_i\}$, that are extensive on the horizon or boundary. The regions $A_i$, $a_i$ and $\mathcal{E}_i$ are mapped to $A_i'$, $a_i'$ and $\mathcal{E}_i'$ respectively. See the right figure in Fig.5 for a time slice of the Rindler bulk. Note that, due to the translation symmetry $a_i'$ and $\mathcal{E}_i'$ are the regions projected to $A_i'$ along the $r$ direction.

The quantum correction to the entanglement entropy (thermal entropy in this case) of $A' = A_1' \cup A_2' \cup A_3'$ is just the bulk entanglement entropy for the region $a'$, which is the exterior region of the Rindler horizon. Since the entanglement contour respects the symmetries, it only depends on the radial coordinate and is flat along the $x_i$ directions, i.e.

$$s_{a'}(\vec{x}, r) = s_{a'}(r). \tag{35}$$

It is convenient to define the constant

$$\mathcal{C} = \int s_{a'}(r) dr, \tag{36}$$

where the domain of the integration is from the horizon to the boundary. Thus $\mathcal{C}$ is the density function for bulk entanglement entropy after integration over the radius direction.

On the boundary $A'$, let us denote the leading order and quantum correction of $S_{A'}$ by $S_{A'}^{(0)}$ and $S_{A'}^{(1)}$ respectively. Due to translation symmetries, $S_{A'}^{(1)}$ and $S_{A'}^{(0)}$ should be equally distributed to all degrees of freedom on $A'$ hence present a volume law. Accordingly, the contour function is a constant given by,

$$s_{A'}(\vec{x}) = \frac{1}{4G_N} + \mathcal{C}, \tag{37}$$

where the first term come from the RT (or Bekenstein-Hawking) formula while the second term come from quantum correction. For a subregion $A_2'$ with length $l_2'$, the PEE $s_{A'}(A_2')$ is just given by

$$s_{A'}(A_2') = l_2' \left( \frac{1}{4G_N} + \mathcal{C} \right). \tag{38}$$

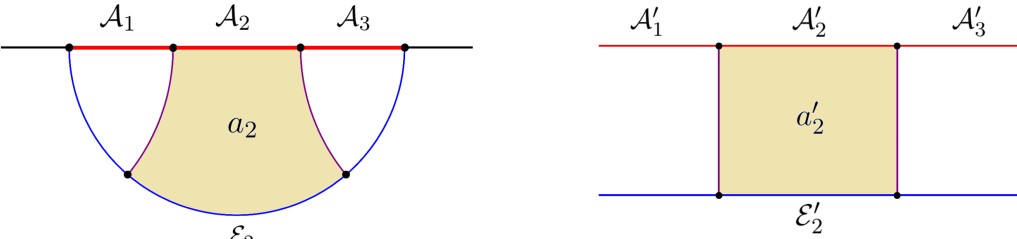

Figure 5: The homology surface $a$ at a time slice is mapped to the time slice of the Rindler bulk under the Rindler transformation. $\gamma_P$ curves are mapped to the curves along the $r$ direction. The bulk region $a_2$ is just mapped to the bulk region $a_2'$, which is the projection region of $A_2'$ along the $r$ direction.

It is easy to see the length of $\mathcal{E}_2'$ equals to $l_2'$, and $s_{a'}(a_2') = l_2' \mathcal{C}$. Then we can write the PEE (38) in the following way,

$$s_{A'}(A_2') = \frac{Area(\mathcal{E}_2')}{4G} + s_{a'}(a_2').$$ (39)

The modular slices in $\mathcal{W}_A$ are just mapped to the $AdS_2$ slices with fixed $\vec{x}$ in the Rindler bulk. Note that the Rindler transformation is also a symmetry of the theory, according to the symmetry property of the PEE we have

$$s_a(a_i) = s_{a'}(a_i'), \qquad s_A(A_i) = s_{A'}(A_i').$$ (40)

The length of the geodesic chords is also invariant under the Rindler transformation,

$$Area(\mathcal{E}_i) = Area(\mathcal{E}_i').$$ (41)

Following (39),(40) and (41), we immediately recover (34).

Our discussion shows that, the quantum correction to the entanglement entropies, PEE or entanglement contour in holographic CFTs are indeed proportional the leading contribution. This is consistent with the quantum result of entanglement contour given in [21].

### 3.3 Quantum correction from the additive linear combination proposal for PEE

Unlike the geometric construction, the *ALC proposal* is not limited to the leading order. In holographic field theories, we can expand the entanglement entropy with respect to $G_N$, i.e.

$$S_A = S_A^{(0)} + S_A^{(1)} + S_A^{(2)} + \cdots,$$ (42)

where $S_A^{(i)}$ is of order $\mathcal{O}(G_N^{i-1})$. Similarly we can expand the PEE in the same way and the *ALC proposal* should hold at all orders, i.e.

$$s_A^{(i)}(A_2) = \frac{1}{2}\left(S_{12}^{(i)} + S_{23}^{(i)} - S_1^{(i)} - S_3^{(i)}\right).$$ (43)

All properties of the PEE should be satisfied respectively at all orders. In the following, we only consider the first order correction, which are the bulk entanglement entropies of $\Sigma_{A_i}$,

$$S_{A_i}^{(1)} = S_{\Sigma_{A_i}}.$$ (44)

So we get another formula for the quantum correction to the PEE

$$s_A^{(1)}(A_2) = \frac{1}{2}\left(S_{\Sigma_{12}} + S_{\Sigma_{23}} - S_{\Sigma_1} - S_{\Sigma_3}\right),$$  (45)

where $\Sigma_i$ means $\Sigma_{A_i}$ and $\Sigma_{ij}$ means $\Sigma_{A_i} \cup \Sigma_{A_j}$.

This looks quite confusing. On the one hand, the linear combination (45) are exactly the same as the *ALC proposal*, hence looks like a bulk PEE $s_{\Sigma_A}(\Sigma_2)$. On the other hand the *ALC proposal* requires $\Sigma_A = \Sigma_1 \cup \Sigma_2 \cup \Sigma_3$ where $\Sigma_i$ are non-overlapping, and furthermore

$$\Sigma_{12} \cap \Sigma_{23} = \Sigma_2, \quad \Sigma_{12} = \Sigma_1 \cup \Sigma_2, \quad \Sigma_{23} = \Sigma_2 \cup \Sigma_3.$$  (46)

Obviously, these requirements are satisfied by the regions $a_i$ rather than the regions $\Sigma_i$ or $\Sigma_{ij}$ in the bulk. So the linear combination in (45) is not a PEE defined by the *ALC proposal* and is not guaranteed to be additive. However, previously we get the result $s_A^{(1)}(A_2) = s_a(a_2)$ using the fine structure analysis of the entanglement wedge. This implies that the left hand side of (45) is a PEE in the bulk thus should be additive.

The confusion can be resolved if one associate the bulk entanglement entropies to their corresponding boundary regions $A_i$ and $A_{ij}$ rather than the bulk regions $\Sigma_i$ and $\Sigma_{ij}$. In other words, the left hand side of (45) can be understand as a PEE on the boundary following the *ALC proposal*. Thus, the additivity immediately follows.

Now we show the additivity of (45) from a more intuitive perspective in the Rindler spacetime. Again we consider the simple case of $AdS_3/CFT_2$ which is shown in Fig.6. Though the RT surfaces for $A_1, A_3, A_1 \cup A_2$ and $A_2 \cup A_3$ look quite different from one another (see the blue solid lines in the upper figure of Fig.6), their images in the Rindler bulk are indeed the same curve up to a translation or reflection (see the solid blue lines in the lower figure of Fig.6), because $A_1', A_3', A_1' \cup A_2'$ and $A_2' \cup A_3'$ are all infinitely long half lines. For example, consider the RT surface emanating from $x = x_0$ and moving along the $+x$ direction, it approaches the horizon in the following way $r(x) = r_h(1 - e^{-(x-x_0)})^{-1}$, where $r = r_h$ is the horizon. In the large $|x|$ region, the RT surfaces just move along the horizon, thus the translation symmetry emerges and the entanglement contour is flat at the large $|x|$ limit. It is obvious that if we translate $\mathcal{E}_{12}'$ by $l_2'$, it exactly matches with $\mathcal{E}_1'$. The only difference is that $\mathcal{E}_{12}'$ is longer by $l_2'$ near the cut off region, where the volume law applies. Also the bulk region $\Sigma_{12}'$ is only larger than $\Sigma_1'$ by a region that exactly matches with $a_2'$ under a translation. Then we have

$$S_{A_1' \cup A_2'}^{(0)} - S_{A_1'}^{(0)} = \frac{l_2'}{4G_N}, \quad S_{A_2' \cup A_3'}^{(0)} - S_{A_3'}^{(0)} = \frac{l_2'}{4G_N},$$  (47)

and

$$S_{\Sigma_{12}'} - S_{\Sigma_1'} = l_2'\mathcal{C}, \quad S_{\Sigma_{23}'} - S_{\Sigma_3'} = l_2'\mathcal{C}.$$  (48)

Plugging the above equations to the *ALC proposal*, as expected at the leading order, we find the PEE is just given by,

$$s_{A'}^{(0)}(A_2') = \frac{l_2'}{4G_N}.$$  (49)

While the quantum correction (45) can also be calculated by

$$\begin{aligned}
s_A^{(1)}(A_2) &= \frac{1}{2}\left(S_{\Sigma_{12}} + S_{\Sigma_{23}} - S_{\Sigma_1} - S_{\Sigma_3}\right) \\
&= \frac{1}{2}\left(S_{\Sigma_{12}'} + S_{\Sigma_{23}'} - S_{\Sigma_1'} - S_{\Sigma_3'}\right) \\
&= l_2'\mathcal{C}.
\end{aligned}$$  (50)

The result recovers the previous result of $s_{a'}(a'_2)$ or $s_a(a_2)$ using the fine structure of the entanglement wedge, so the additivity of the right hand side of (45) is justified in this case. This is also a consistency check between the two approaches to evaluate the quantum corrections.

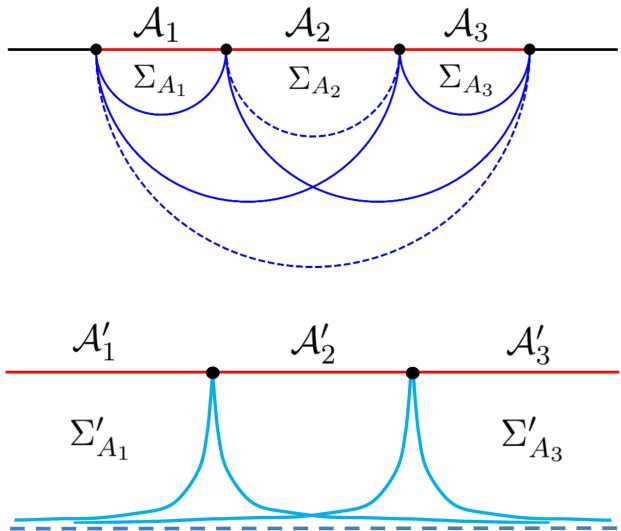

Figure 6: The upper figure shows the RT curves (solid blue curves) for the subregions in the linear combination in the *ALC proposal*. The lower figure shows the images of the above RT curves in the Rindler spacetime. Since $A'_1, A'_3, A'_1 \cup A'_2$ and $A'_2 \cup A'_3$ are all infinite half lines, their RT surfaces are the same up to a translation or reflection.

## 4 Discussion

In this paper, we explore two important aspects about entanglement contour. Firstly we explore the "first law" of entanglement contour. For a given region, the first law tells us that first order variation of the contour (or density) function of the entanglement entropy equals the first order variation of the expectation value of the contour (or density) function of the modular Hamiltonian, i.e. $\delta s_A(\mathbf{x}) = \delta \langle k_A(\mathbf{x}) \rangle$. This gives a much stronger and finer description of the variation of the entanglement structure under the variation of the state. Note that, this relation is only derived for the configurations where the *ALC proposal* applies. However this relation seems to be a quite natural extension of the first law of entanglement entropy $\delta S_A = \delta \langle K_A \rangle$. We conjecture it to be true for more generic configurations. It may be quite useful to calculate the entanglement contour for low energy exited states (see for example [42]).

The second aspect is the quantum correction to the holographic entanglement contour. Firstly, using the fine structure picture, we find that the quantum correction to the PEE of a subset is captured by the bulk PEE of a certain bulk subregion inside the homology surface, i.e. $s_A^{(1)}(A_2) = s_a(a_2)$. This observation gives a fine relation between the PEE of the bulk degrees of freedom and the PEE of the boundary degrees of freedom. Secondly, for the configurations where the *ALC proposal* applies, the quantum corrections to the PEE computed by the *ALC proposal* is a linear combination of the bulk entanglement entropies of certain bulk regions. For example, see the right hand side of (45). The results of the two approaches are confirmed to be consistent in the Rindler bulk spacetime. Note also that, the additivity of the linear combination (45) is not manifest. It comes from the additivity of the boundary PEE and the fact

that the bulk entanglement entropies are quantum corrections to the entanglement entropies of certain boundary regions.

One can test the first law of the entanglement contour at the leading order using a perturbed geometry around the pure AdS space. On one hand, the perturbation of the geometry perturbs the stress tensor of the boundary CFT, which furthermore perturbs the entanglement contour according to the first law. On the other hand, the perturbation of the geometry perturbs the fine correspondence between points in $A$ and $\mathcal{E}_A$ which also gives a perturbation of the entanglement contour. The first law can be confirmed if the two perturbations of the entanglement contour coincide with each other.

Explicit configurations of bit threads is a good way to describe the entanglement contour. However, the entanglement contour is assumed to be unique while the bit thread configuration is highly non-unique even when the state and region are determined. So far, it is not well undertood how we can impose physical requirements to determine the bit thread configuration for a given entanglment wedge. We propose that, reproducing the right entanglement contour should be a reasonable physical requirement. This is recently explored in [36] by applying the locking theorems [58, 59] of bit threads to construct a concrete locking scheme for the RT surfaces in the entanglement wedge. In [60] two perturbations of the bit threads configurations are explicitly considered. One of them is for the geodesic bit threads normal to the RT surface [57], consistent with our fine structure analysis [29][9]. The other is the canonical perturbation of the bit threads configuration following the Iyer-Wald formalism [63]. These perturbations of bit threads give perturbations of the entanglement contour, hence is useful to test the first law.

We do not explicitly discuss the dependence of the coordinates of the bulk entanglement contour $s_a(\mathbf{x})$. It is interesting since it gives a fine description of the entanglement structure in the bulk and affects the boundary entanglement contour at the quantum level. The bulk entanglement contour is also mentioned recently in [34, 35], which extend the concept of bit threads to the quantum bit threads by allowing the bit threads to start and terminate in the bulk. In such a way they can use the quantum bit threads to describe the quantum correction of the holographic entanglement entropy. However an explicate configuration of the quantum bit threads is necessary to give a contour function.

We propose that the entanglement contour $s_a(\mathbf{x})$ should be evaluated by applying the first law of entanglement contour in the bulk. More explicitly let us consider the low energy excitations (for example the Hawking Radiation) in the bulk which induce a perturbation of the stress tensor, while the backreaction to the geometry can be omitted. According to the first law of entanglement contour, the perturbation of the contour function is proportional to the perturbation of the stress tensor. This approach may be valid at the early age of a black hole. Together with the picture we give in section 3.2, the time evolution or perturbation of the bulk entanglement contour is furthermore related to the evolution or perturbation of the boundary entanglement contour at the quantum level using the relation between the bulk PEE and boundary PEE. See Fig.7 for example.

Another important relevant question that we come up with is what the first law of the entanglement contour at the quantum level can tell us about the dynamics in the bulk. The linearized Einstein's equations in the bulk have already been derived by the first law of the entanglement entropy at the classical level. According to our discussion on both of the first law and quantum correction of the entanglement contour, the perturbation of the boundary entanglement contour at the quantum level should relate to the perturbation of the energy-momentum tensor in the bulk, and further relate to the perturbation of the bulk geometry. We hope this can give us further understanding about the dynamics of geometry beyond the

---

[9]See also [61, 62] for another related flow picture based on a fracton model which satisfies several major properties of AdS/CFT.

linearized Einstein's equations or quantum excitations of gravity.

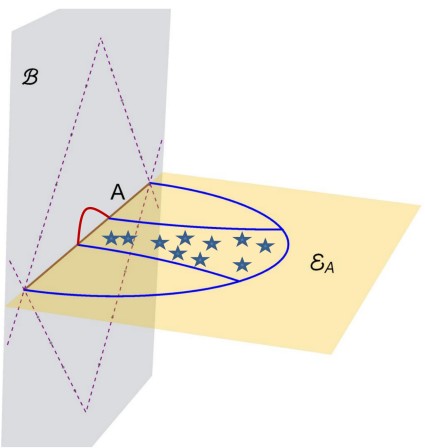

Figure 7: Here the homology surface $\Sigma_A$ is divided into $a_1 \cup a_2 \cup a_3$ as in Fig.4. The background state is the vacuum of the boundary CFT, and the stars are low energy excitations of the stress tensor inside $a_2$. The red curve on the boundary is the perturbation of the entanglement contour at the quantum level caused by the bulk excitations, which is only non-zero on $A_2$.

# Acknowledgments

We would like to thank Andrew Rolph and Juan Pedraza for a careful reading of this manuscript and helpful comments. QW is supported by the "Zhishan"Scholars Programs of Southeast University. MH receives support from the National Science Foundation through grant PHY-1912278.

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
