# Peer review of "First Law and Quantum Correction for Holographic Entanglement Contour"

_SciPost Physics, doi:SciPost Phys. 11, 058 (2021)_

## Round 1 · Referee Report · Anonymous (Referee 1) · 2021-8-2

Strengths

1-Well structured
2-Clear motivation
3-Active topic of research

Weaknesses

1-Lack of unique definition of entanglement contour
2-Non-locality of partial modular Hamiltonian and the modular Hamiltonian density seems hidden away
3-A few typos and grammatical errors

Report

This article contains interesting insights into the properties of two definitions of the entanglement contour. The main issue with this research direction is the lack of a unique definition of entanglement contour. As the authors mention, there is hope that because some of these properties are consistent across definitions, that they are indeed robust to the particular definition of entanglement contour. This is an interesting addition to the program of studying the relation between entanglement and geometry which is a very active field of research.

The article open a new pathway in an existing research direction and satisfies all the general acceptance criteria. Therefore I would recommend publication and have added a few comments and recommendations.

Requested changes

1-Eqs. (2.13), (2.14) and (2.16) seem to imply a local contour function k_A(x) defined on x. Looking at eq. (2.10) this seems to be far from the case. Perhaps a comment clearly stating whether k_A(x) is a local operator or not would help with clarity. There are some examples in the literature where the modular Hamiltonian is local and others in which it is not. Is there a connection between the local Hamiltonian density and the contour function k_A(x) in either of those cases?

2-In the (warped) AdS/(warped) CFT correspondence and 3-dimensional flat holography examples mentioned in the introduction, is there a connection between the non-Lorentz invariance of the dual field theory and the RT surfaces not being anchored on the boundary? If this is the case it would be interesting to add as a comment or footnote.

3-In section 3.2.2, the definition of holographic entanglement contour seems to rely on the boundary subregion and the RT surface having a clear one to one mapping. This seems natural when the subregion is a single interval of a pure state. Is there a standard way to deal with two intervals in the connected phase where this seems less obvious? Similarly, what about for a black hole geometry when the interval is large enough that the RT surface includes a horizon piece? Similar to point 2, it would be interesting to discuss this as a comment or a footnote.

4-Equation (2.4) seems to have a sign mistake. Before eq. (3.8), Rindler is misspelled.

  • validity: good
  • significance: good
  • originality: good
  • clarity: good
  • formatting: good
  • grammar: good

Author:  Qiang Wen  on 2021-08-19  [id 1693]

(in reply to Report 1 on 2021-08-02)

We thank the referee very much for his or her recommendation.

The uniqueness of the entanglement contour is a foundational problem of this direction. Consider that fact that all the results we got are highly consistent, the existance of additive structures and the arguments for the uniqueness for Poincare invariant theories, we feel optimistic for this problem. The concept of entanglement contour should be studied more deeply based on the density matrix. We did another proof reading to fix typos and grammatical errors.

comment 1
The contour $k_A(x)$ is not a local function since is also depends on the region A. we added the footnote 5 to clarify this. Also we add a paragraph at the end of section 2 to clarify that writting $K_A$ as a integral of $k_A(x)$ doesnot rely on the existence of the local Hamiltonian density. It only relies of the additivity of the partial modular Hamiltonian.

comment 2
This could be interesting for those who are not familiar with the holography beyond AdS/CFT. I add the footnote 1 to further explain this point.
When the boundary field theory is non-Lorentz invariant, the causal development of, for example, an interval becomes an infinitely long strip instead of a causal diamond. In order to keep the consistency between the bulk and boundary causal structure, the extremal surface should not be anchored on the boundary. This is discussed in detail in one of my previous publication.

comment 3
We add a sentence in footnote 7 to clarify this point. Thanks for pointing out that.
However, similar construction cannot be straitforwardly generalized to the cases of multi-intervals and a large enough boundary interval in the BTZ background with disconnected RT surface, since the modular flow becomes nonlocal. This is an important problem we hope to understand further in the future.

Indeed any discussion on the entanglement for disconnected regions will be highly involved. We hope we can make progress in this direction.

comment 4
Thanks very much for pointing out that. This is a vital typo and we fixed it.

---

## Round 1 · Referee Report · Pablo Bueno (Referee 2) · 2021-8-16

Strengths

1- Reasonably well written and motivated.

2- The ALC approach is interesting and its use in deriving the first law of entanglement contour is smart.

3- The proposal for the quantum correction to the entanglement contour for holographic entanglement entropy seems reasonable (at least for highly symmetric regions).

Weaknesses

1- It would be desirable to have a proposal which is valid for general regions (as opposed to the ALC one, which only works for very particular cases). It is not clear to me the new first law they derive would work straightforwardly in general.

1- The meaning of 2.13 and 2.14 is a bit unclear given that the modular Hamiltonian cannot be written as an integral over the entangling regions in most cases (see report).

Report

The authors study the possibility of rewriting the first law of entanglement entropy (which, for a given state, relates its variation to a nearby state to the variation of the modular Hamiltonian expectation value) in terms of the entanglement contour. Given some entangling region A, the entanglement contour is defined as a sort of a density of EE: for a subregion A_i of A it is defined as the integral of a certain density over such subregion A_i; when integrated over the whole A, this gives the EE. In order to derive a first law for the entanglement contour, the authors use a smart proposal (which they call ALC from ``additive linear combination’’). This only works for regions which can be partitioned into three non-overlapping regions which can be ordered (this includes intervals in two dimensions, strips and spheres in higher dimensions). It consists in writing the entanglement contour for a given subregion as a linear combination of entanglement entropies involving the three subregions of the partition. This rewriting allows the authors to write the variation of the entanglement contour in terms of variations of EE's of different subregions, which can be in turn related to variations of the modular Hamiltonian via the first law. Defining a notion of density for the modular Hamiltonian as well, the authors are finally led to a first law of entanglement contour which would imply the well known first law of entanglement entropy. In the second half of the paper, the authors try to come up with a characterization of the leading quantum correction to the holographic entanglement contour (for the holographic EE this is given by the bulk entanglement entropy of the region encircled by the Ryu-Takayanagi surface). The authors propose that the corresponding contribution to the holographic entanglement contour for a boundary subregion is given by the entanglement contour of certain bulk subregion inside the Ryu-Takayanagi surface and perform some checks in highly symmetric situations.

I think the paper contains some interesting material which deserves publication. I only have a question which I think the authors should at least briefly comment on before accepting the paper. In the paper the authors restrict their analysis to highly symmetric regions, for which the ALC proposal can be applied. In the case of spherical regions, it is also the case that the modular Hamiltonian is local for general CFTs and it therefore can be written as an integral over the entangling region A. The authors mention this case as further support for their definitions of the partial modular Hamiltonian. However, for most entangling regions, it is not possible to write the modular Hamiltonian as an integral of that kind (spheres are an exception, not the norm). Hence, I am a bit worried about the validity of equations like 2.13 and 2.14 on more general grounds. I think the authors should clarify the meaning of those equations in view of this.
  • validity: good
  • significance: good
  • originality: high
  • clarity: good
  • formatting: good
  • grammar: reasonable

Author:  Qiang Wen  on 2021-08-19  [id 1692]

(in reply to Report 2 by Pablo Bueno on 2021-08-16)
Category:
answer to question

We thank Dr. Pablo Bueno very much for his recommendation.

We understand you concern in more generic configurations where the modular Hamiltonian cannot be written as an integration of the stress tensor. We stress that, the reason we can write $K_A$ as an integral of $k_A(\textbf{x})$ is the additivity of the partial modular Hamiltonian we constructed. This only relies only on the validaty of the ALC proposal. There are many cases where the ALC proposal applies while the $K_A$ can not be written as a integral of the stress tensor. Because the ALC proposal does not select theories. For example, let us consider an interval $A$ in a 2-dimension theory which is not conformal invariant. Since in this case the ALC proposal applies thus the partial modular Hamiltonian is additive, $K_A$ can still be written as a integral of $k_A(\textbf{x})$ with $k_A(\textbf{x})$ not directly related to the stress tensor.

We add a paragraph in the end of section 2 to clarify this point.

---

## Round 2 · Referee Report · Pablo Bueno (Referee 2) · 2021-8-30

Report

I think the authors have properly addressed my concerns and the paper is now ready for publication.

---

## Round 2 · Referee Report · Anonymous (Referee 1) · 2021-9-3

Report

The authors have addressed my questions and comments. The article fits the criteria for publication.

---

## Round 2 · Author Response

We thank the referees for recommendation and a careful reading our manuscript. Also thanks very much for the valuable comments and suggestions. In the revised manuscript, we made the corrections for further clarification following the referee's comments.

---

## Round 2 · List of Changes

We improved the presentation of the manuscript, fixed the typos and grammatical errors we can find.

we add 4 references. These includes the interpretation of the ALC proposal as a conditional mutual information (page4) and mention that the entanglement contour has been applied to the island formla to depict the entanglement structure of the hawking radiation (page3).

we added the footnote 1, 5, 7 in accordance with the comments or suggestions from the referees.

we added a paragraph in the end of section 2 to explain a concern from both of the referees.

---

## Editorial Decision

published